

# The genetics of venom ontogeny in the eastern diamondback rattlesnake (*Crotalus adamanteus*)

Darin R. Rokyta[1], Mark J. Margres[1,2], Micaiah J. Ward[1] and Elda E. Sanchez[3,4]

[1] Department of Biological Science, Florida State University, Tallahassee, FL, United States of America
[2] School of Biological Sciences, Washington State University, Pullman, WA, United States of America
[3] Department of Chemistry, Texas A&M University—Kingsville, Kingsville, TX, United States of America
[4] National Natural Toxins Research Center, Texas A&M University—Kingsville, Kingsville, TX, United States of America

## ABSTRACT

The same selective forces that give rise to rapid inter- and intraspecific divergence in snake venoms can also favor differences in venoms across life-history stages. Ontogenetic changes in venom composition are well known and widespread in snakes but have not been investigated to the level of unambiguously identifying the specific loci involved. The eastern diamondback rattlesnake was previously shown to undergo an ontogenetic shift in venom composition at sexual maturity, and this shift accounted for more venom variation than geography. To characterize the genetics underlying the ontogenetic venom compositional change in *C. adamanteus*, we sequenced adult/juvenile pairs of venom-gland transcriptomes from five populations previously shown to have different adult venom compositions. We identified a total of 59 putative toxin transcripts for C. adamanteus, and 12 of these were involved in the ontogenetic change. Three toxins were downregulated, and nine were upregulated in adults relative to juveniles. Adults and juveniles expressed similar total levels of snake-venom metalloproteinases but differed substantially in their featured paralogs, and adults expressed higher levels of Bradykinin-potentiating and C-type natriuretic peptides, nerve growth factor, and specific paralogs of phospholipases $A_2$ and snake venom serine proteinases. Juvenile venom was more toxic to mice, indicating that the expression differences resulted in a phenotypically, and therefore potentially ecologically, significant difference in venom function. We also showed that adult and juvenile venom-gland transcriptomes for a species with known ontogenetic venom variation were equally effective at individually providing a full characterization of the venom genes of a species but that any particular individual was likely to lack several toxins in their transcriptome. A full characterization of a species' venom-gene complement therefore requires sequencing more than one individual, although the ages of the individuals are unimportant.

Corresponding author
Darin R. Rokyta,
drokyta@bio.fsu.edu,
drokyta@gmail.com

## INTRODUCTION

Snake venoms are traits of moderate genetic complexity comprised largely of proteinaceous toxins that function in predation and defense (*Boldrini-França et al., 2010*; *Calvete et al.,*

*2010*; *Rokyta et al., 2011*; *Rokyta et al., 2012*; *Durban et al., 2013*; *Margres et al., 2013*). With some exceptions (*Margres et al., 2015a*; *Margres et al., 2016b*), snake venoms have been found to evolve rapidly under positive selection within and between species, involving both changes in toxin expression patterns (*Gibbs, Sanz & Calvete, 2009*; *Rokyta et al., 2015*; *Margres et al., 2015a*; *Margres et al., 2015b*) and protein sequences (*Lynch, 2007*; *Gibbs & Rossiter, 2008*). This rapid evolution is thought to result from the evolutionarily critical roles of venom in feeding and defense (*Jansa & Voss, 2011*) and the antagonistic coevolutionary interactions with predators and prey (*Biardi, Chien & Coss, 2005*; *Biardi et al., 2011*). The same selective pressures that result in local adaptation and species-level divergence in venoms, however, can also operate on life-history stages, particularly if prey or predators change with age or size. Ontogenetic changes in venom composition have been identified in numerous snake species (e.g., *Mackessy, 1988*; *Glenn, Straight & Wolf, 1994*; *López-Lozano et al., 2002*; *Saldarriaga et al., 2003*; *Mackessy et al., 2006*; *Calvete et al., 2010*; *Zelanis et al., 2010*; *Durban et al., 2013*; *Rokyta et al., 2015*), but certainly not in all species that have been examined (e.g., *Gibbs et al., 2011*; *Rokyta et al., 2015*).

The eastern diamondback rattlesnake (*Crotalus adamanteus*) is native to the southeastern United States and is the largest rattlesnake species. *Crotalus adamanteus* specializes on endothermic prey, with rats, squirrels, and rabbits comprising the majority of its diet (*Klauber, 1997*), and has one of the most well-characterized venom-gland transcriptomes (*Rokyta et al., 2011*; *Rokyta et al., 2012*; *Rokyta, Margres & Calvin, 2015*) and venom proteomes of any snake species (*Margres et al., 2014*; *Eichberg et al., 2015*). *Margres et al. (2015a)* detected significant interpopulation variation in venom composition for *C. adamanteus*, and *Margres et al. (2015b)* showed that the venom of *C. adamanteus* undergoes an ontogenetic shift at sexual maturity and that this shift accounts for more of the total venom compositional variation than geography. *Wray et al. (2015)* also detected a significant change in venom composition between birth and the first postnatal shed for *C. adamanteus*. The reference transcriptomes used in these proteomic characterizations were, however, all derived from a single juvenile, and none of these approaches were able to unambiguously identify the loci contributing to these forms of intrapopulation and intraindividual variation.

We investigated the genetics and transcriptomics of the venom ontogenetic change in *C. adamanteus* by comparing venom-gland transcriptomes across pairs of adults and juveniles from five populations previously determined to show different adult venom compositions (*Margres et al., 2015b*). We conducted the first RNA-seq based comparison of the venom-gland transcriptomes of adults and juveniles of a snake species with adequate replicates for statistical comparisons. Although many studies have demonstrated the presence of an ontogenetic change in venom composition and have even identified the types of toxins involved, none have resolved the nature of the change to particular paralogs within the context of a complete species-level characterization of venom composition that incorporates data from individuals from throughout the entire range of a species. We also addressed methodological issues in venom-gland transcriptomics, including repeatability, the utility of pseudobiological replicates (i.e., separate glands for the same individual), and

**Table 1 Specimen information.** Population designations follow Fig. 1.

| Name | ID | Population | Age class | SVL (cm) | TL (cm) | Sex |
|------|-----|-----------|-----------|----------|---------|-----|
| ANF-A | KW1264 | Apalachicola National Forest | Adult | 128.0 | 137.0 | Female |
| ANF-J | MM0198 | Apalachicola National Forest | Juvenile | 67.0 | 71.0 | Female |
| BR-A | MM0127 | Brooksville Ridge | Adult | 143.0 | 157.5 | Male |
| BR-J | KW2171 | Brooksville Ridge | Juvenile | 64.0 | 69.0 | Female |
| CAL-A | KW1942 | Caladesi Island | Adult | 165.0 | 180.0 | Male |
| CAL-J | KW2170 | Caladesi Island | Juvenile | 41.5 | 44.5 | Male |
| ENP-A | KW0944 | Everglades National Park | Adult | 117.5 | 130.0 | Male |
| ENP-J | MM0143 | Everglades National Park | Juvenile | 88.0 | 96.0 | Male |
| LSG-A | KW2161 | Little St. George Island | Adult | 129.5 | 142.5 | Male |
| LSG-J | KW2184 | Little St. George Island | Juvenile | 78.0 | 83.0 | Female |

**Notes.**
Abbreviations: SVL, snout-vent length; TL, total length.

the consequences of using single animals to estimate the full complement of venom genes for an entire species.

## METHODS

### Animals and tissues

Adult/juvenile pairs of *C. adamanteus* were collected from five populations previously shown to have different venom compositions (Fig. 1; *Margres et al., 2015a*; *Margres et al., 2015b*). Individuals were collected live from the field and temporarily housed at Florida State University; animal information is provided in Table 1. Maturity was assessed on the basis of snout-vent length (SVL), with individuals ≥102.0 cm classified as adults and individuals <102.0 cm classified as juveniles. *Waldron et al. (2013)* used SVL data to estimate maturity in *C. adamanteus* and determined that 102.0 cm SVL was the smallest size of reproductively active individuals, and *Margres et al. (2015b)* found that the ontogenetic shift in toxin gene expression in *C. adamanteus* corresponded with sexual maturation (i.e., occurred at approximately 1 m SVL). We followed the approach of *Rokyta et al. (2012)* for preparation of venom glands. Briefly, we stimulated transcription in the glands by means of venom extraction under anesthesia (*McCleary & Heard, 2010*). Each individual was anesthetized with a propofol injection (10 mg/kg), and venom expulsion was initiated by means of electrostimulation. After allowing four days for transcription to reach maximum levels (*Rotenberg, Bamberger & Kochva, 1971*), each individual was euthanized by injection of sodium pentobarbital (100 mg/kg). Left and right venom glands were removed and transferred into separate aliquots of RNAlater. Specimens were collected under the following permits: Florida Fish and Wildlife Conservation Commission (FWC) LSSC-13-00004 and LSSC-09-0399 and Florida Department of Environmental Protection permit #03131424. The above procedures were approved by the Florida State University Institutional Animal Care and Use Committee (IACUC) under protocol #0924.

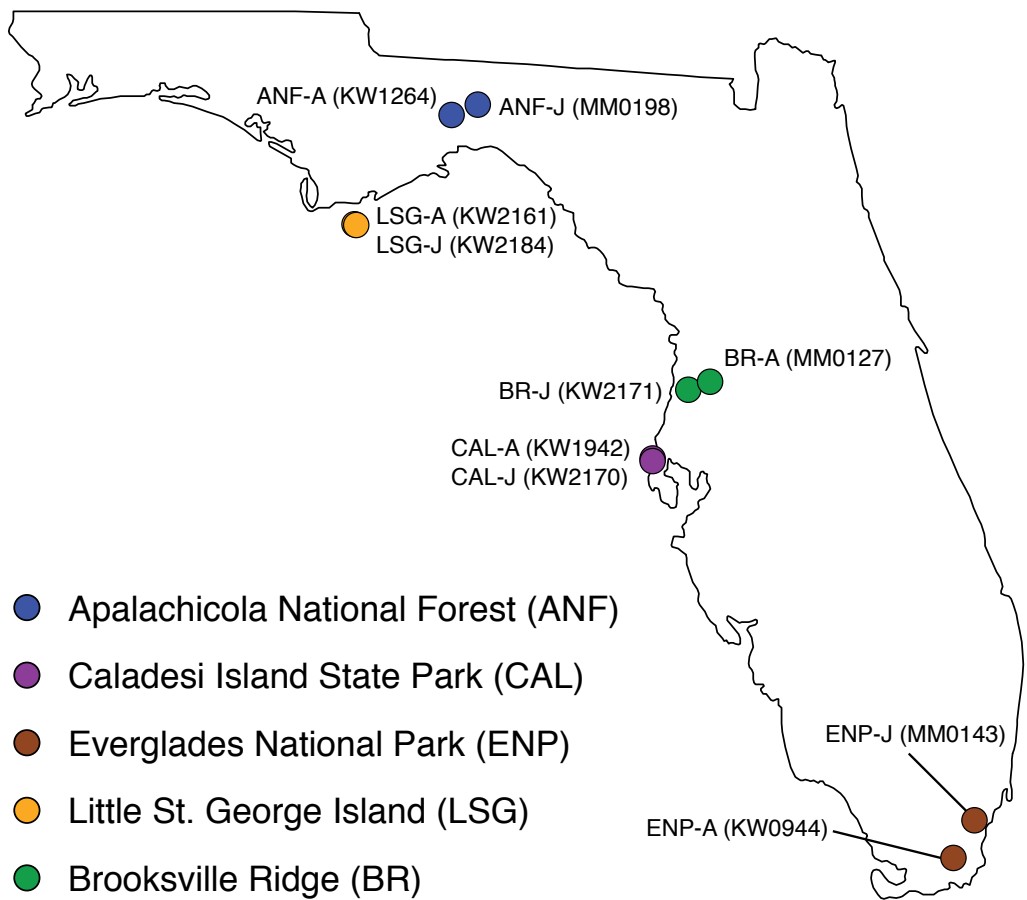

**Figure 1** **Collection localities of ten *Crotalus adamanteus* individuals used for venom-gland transcriptomics.** Adult/juvenile pairs were collected from five regions previously shown (*Margres et al., 2015a*) to have different adult venom compositions. Abbreviations: A, adult; J, juvenile.

## Venom-gland transcriptome sequencing

Snake venom-gland RNA extraction was performed as previously described (*Rokyta et al., 2011*). Briefly, venom-gland RNA was extracted by first homogenizing the venom gland tissue and submerging in Trizol (Invitrogen), followed by the addition of 20% chloroform and centrifugation in phase lock heavy gel tubes (5Prime) to separate the RNA from DNA and other cellular debris. Isolated RNA was then pelleted with isopropyl alcohol and washed with 75% ethanol. A secondary ethanol precipitation step was used to further purify and concentrate the RNA by using Pellet Paint Co-Precipitant (EMD Millipore), 10% sodium acetate, and 100% ethanol, followed by centrifugation. Purified RNA was then washed with 70% ethanol and quality checked using a Bioanalyzer with an RNA 6000 Pico Kit (Agilent Technologies) following the manufacturer's instructions.

Prior to RNA-seq library preparation, 1–2 µg of total RNA was used to isolate mRNA using the NEBNext Poly(A) mRNA Magnetic Isolation Module (New England Biolabs). A fragmentation time of 15.5 min was used to achieve fragment sizes of approximately 370 nucleotides (adapter-ligated). The purified mRNA was immediately used for cDNA library

**Table 2** Summary of sequencing and read merging.

| Name | Gland | Sample ID | Read pairs | Merged reads | Average merged length |
|------|-------|-----------|-----------|--------------|----------------------|
| ANF-A | Left | KW1264-L | 8,183,782 | 6,840,866 | 184 |
| | Right | KW1264-R | 8,813,501 | 7,646,296 | 183 |
| ANF-J | Left | MM0198-L | 10,093,280 | 8,665,819 | 185 |
| | Right | MM0198-R | 12,198,682 | 10,689,006 | 167 |
| BR-A | Left | MM0127-L | 4,859,835 | 4,080,422 | 178 |
| | Right | MM0127-R | 8,638,140 | 7,551,717 | 179 |
| BR-J | Left | KW2171-L | 7,082,608 | 6,161,356 | 178 |
| | Right | KW2171-R | 8,780,220 | 7,574,258 | 183 |
| CAL-A | Left | KW1942-L | 9,779,779 | 8,145,813 | 182 |
| | Right | KW1942-R | 13,789,122 | 11,335,770 | 186 |
| CAL-J | Left | KW2170-L | 6,589,939 | 5,827,859 | 179 |
| | Right | KW2170-R | 4,168,769 | 3,727,631 | 177 |
| ENP-A | Left | KW0944-L | 6,567,272 | 5,568,110 | 166 |
| | Right | KW0944-R | 6,729,304 | 5,893,204 | 157 |
| ENP-J | Left | MM0143-L | 11,387,670 | 10,132,636 | 174 |
| | Right | MM0143-R | 11,474,669 | 10,314,837 | 172 |
| LSG-A | Left | KW2161-L | 8,167,858 | 7,100,692 | 176 |
| | Right | KW2161-R | 11,332,465 | 9,434,016 | 175 |
| LSG-J | Left | KW2184-L | 10,226,589 | 8,680,851 | 180 |
| | Right | KW2184-R | 1,863,304 | 1,604,848 | 172 |

preparation using the NEBNext Ultra RNA Library Prep Kit and Multiplex Oligos for Illumina (New England Biolabs). PCR was performed using the NEBNext High-Fidelity 2X Hot Start PCR Master Mix and 14 cycles of PCR to achieve the desired DNA concentration for sequencing. DNA was purified using Agencourt AMPure XP PCR Purification Beads throughout and at the end of the protocol. The library quality was assessed using a Bioanalyzer with a High Sensitivity DNA Kit (Agilent Technologies) following the manufacturer's instructions. To determine the amplifiable concentration, KAPA PCR was performed on individual samples by the Florida State University Molecular Cloning Facility. Using the amplifiable concentrations, individual samples with unique indices were pooled to achieve the desired final concentration for sequencing such that each individual snake venom gland was equally represented. The quality of the pooled DNA samples was assessed using a Bioanalyzer with a High Sensitivity DNA Kit (Agilent Technologies), and an additional round of KAPA PCR was performed to confirm amplifiable concentration of the pooled sample prior to sequencing.

All sequencing was performed on an Illumina HiSeq 2000 by the Florida State University College of Medicine Translational Science Laboratory. Samples were multiplexed and sequenced in a rapid run with 150 nucleotide paired-end reads. Left and right glands were processed and sequenced separately for each individual as technical and pseudobiological replicates. The numbers of read pairs per sample are provided in Table 2.

## Transcriptome assembly and analysis

Sequencing reads passing the Illumina quality filter were merged on the basis of their 3′ overlaps with PEAR version 0.9.6 (*Zhang et al., 2014*). Merged reads from both glands for each individual (≤10 million total) were assembled with SeqMan NGen version 12.3.1 with default *de novo* transcriptome settings, retaining only contigs with ≥200 assembled reads. In addition, 1,000 merged reads for each individual were used as seeds with our in-house assembler Extender (*Rokyta et al., 2012*), which simply extends the ends of provided seeds on the basis of reads that overlap and match these ends. We only used merged reads as seeds or for extension if all positions had phred quality scores ≥30. We required an exact match of 120 nucleotides for extension. This second *de novo* assembler was included because of its superior performance at assembling long toxins with high levels of paralogy such as snake-venom metalloproteinases. Note that more commonly used assemblers such as Trinity perform poorly for snake venom transcriptomes (*Rokyta et al., 2012*; *Archer et al., 2014*). Toxin-encoding transcripts were identified by means of blastx (v.2.2.30+) searches with a minimum e-value of $10^{-4}$ against the UniProt animal venom proteins and toxins database (http://www.uniprot.org/program/Toxins) downloaded on November 16, 2015. For each assembly, we retained only full-length putative toxins. We combined the results from both assemblies for each of the ten individuals, eliminated duplicates, and then screened for chimeric sequences by aligning all merged reads from both glands against the unique putative toxin coding sequences with bowtie2 version 2.2.7 (*Langmead & Salzberg, 2012*). Transcripts showing strongly multimodal or extremely uneven coverage distributions were eliminated, and the remaining transcripts were clustered into groups within individuals showing ≤1% nucleotide divergence. A consensus toxin transcriptome across all ten individuals was generated in Geneious version 8.1.8. Putative toxin-transcript coding sequences were aligned by gene family using the ClustalW algorithm (*Thompson, Higgins & Gibson, 1994*). Transcripts with ≤1.5% nucleotide divergence were combined into clusters by taking the consensus sequence of all cluster members. This clustering of sequences provided an operational definition of paralogs in our final transcriptome. Nontoxin transcripts were derived from a previously published venom-gland transcriptome assembly for *C. adamanteus* (*Margres et al., 2015a*). We used RSEM version 1.2.28 (*Li & Dewey, 2011*) with bowtie version 1.1.2 (*Langmead et al., 2009*) as the aligner to estimate transcript abundances.

To determine the repeatability of our RNA-seq protocol and transcript-abundance estimates, we aligned left- and right-gland merged reads of each individual separately against the consensus transcriptome and used the estimates of transcripts per million (TPM) from RSEM as our abundance estimates, following the approach of *Rokyta, Margres & Calvin (2015)*. Any estimates of 0.0 TPM were replaced with a value of 1.0 TPM to allow logratio transforms necessary for comparisons across glands for individuals.

To test for the absence of consensus transcripts in any of the ten sequenced transcriptomes, we aligned all of the merged reads from the left and right venom glands from each individual against the consensus transcriptome with BWA MEM (https://sourceforge.net/projects/bio-bwa/), using the -M option. Individual reads that showed more than two mismatches (gaps or nucleotide differences) were removed from

the alignments, and we used picard (http://broadinstitute.github.io/picard/) for sorting and indexing. We then used bedtools (*Quinlan & Hall, 2010*) to calculate the coverage for each site of each of the 59 consensus toxin-encoding transcripts. Transcripts with more than 10% of the coding sequence showing less than $5\times$ coverage were considered to be absent from the transcriptome. Note that we applied a strict criterion to consider a transcript as present in a transcriptome for this analysis, which was meant to approximate its potential for being assembled *de novo*. In all other abundance-based analyses, this criterion was not applied. Some transcripts identified as absent for this analysis, therefore, still have abundance estimates for other analyses. To assess the effects of differences in numbers of reads among samples for this analysis, we repeated the analysis using only 9.5 million merged reads for each sample.

To test for evidence of differential expression for toxin genes and to concomitantly account for geographic expression variation, we implemented a pairwise test on adult/juvenile pairs from each population. Because we focused on toxin expression variation, we used available nontoxin expression levels to estimate a null distribution of adult/juvenile expression divergence, then looked for toxins that were outliers relative to this null distribution. Expression levels were estimated as transcripts per million (TPM) using RSEM version 1.2.28 (*Li & Dewey, 2011*) with bowtie version 1.1.2 (*Langmead et al., 2009*) as the aligner and centered logratio (clr) transformed. To generate null distributions, we took the absolute values of the difference in transformed adult and juvenile expression levels for each nontoxin transcript and found the value for the 99th percentile. Toxin transcripts showing differences larger than the nontoxin-based 99th percentile were considered outliers for that particular adult/juvenile pair. Because of the known geographic variation for *C. adamanteus*, we only considered a toxin to show ontogenetic variation if it was an outlier for a majority of the comparisons ($\geq 3$ of 5 comparisons) and the change was in the same direction for a majority ($\geq 3$ of 5 comparisons) of comparisons (i.e., either upregulated or downregulated in adults). We also tested for differential expression using DESeq version 1.26.0 (*Anders & Huber, 2010*) and DESeq2 version 1.14.1 (*Love, Huber & Anders, 2014*), using a 0.1 false-discovery rate (FDR) threshold for both.

## Toxicity assays

To generate a pool of adult venoms, we combined 10 mg per individual for 10 adult snakes from the Apalachicola National Forest (ANF). For the juvenile venom pool, we combined 5 mg per individual for 13 juvenile snakes from the ANF population. Five groups of eight mice per venom were housed in cages and observed throughout the experiments. The median lethal doses (LD$_{50}$s) of snake venoms were determined in BALB/c mice. Venoms were dissolved in 0.85% saline at the highest concentration of venom that was used for the injection (4.89 mg/kg in adult venom and 15.79 mg/kg in juvenile venom). Two-fold serial dilutions using saline were made to obtain four additional concentrations. All solutions during the experiment were stored at 4 °C and warmed to 37 °C just before being injected into mice. Lethal toxicity was determined by injecting 0.2 mL of venom (at various concentrations) into the tail veins of 18–20 g BALB/c mice. The injections were administered using a 1 mL syringe fitted with a 30-gauge, 0.5-in. needle. A saline control

was used. The endpoint of lethality of the mice was determined after 48 h. Calculations of final $LD_{50}$s were determined by the Spearman-Karber method (*Spearman-Karber, 1964*).

## RESULTS AND DISCUSSION

### A complete consensus venom-gland transcriptome for *Crotalus adamanteus*

We separately assembled and annotated the toxin-encoding genes from the venom-gland transcriptomes (Table 2) from ten individuals (five adults and five juveniles) of *C. adamanteus* and merged them into a single representative consensus transcriptome for the species. Our final set of transcripts consisted of 59 putative toxin-encoding transcripts. The venom-gland transcriptome of *C. adamanteus* was previously characterized by means of 454 (*Rokyta et al., 2011*) and 100-nucleotide paired-end Illumina sequencing (*Rokyta et al., 2012*; *Rokyta, Margres & Calvin, 2015*; *Margres et al., 2015a*). These previous characterizations, however, were all based on RNA from the venom glands of the same single juvenile female from the Apalachicola National Forest (ANF). The most recent assembly (*Margres et al., 2015a*) identified 44 toxin clusters for *C. adamanteus*. We have substantially increased this number through a combination of longer reads, which helps the assembly of longer toxins such as snake-venom metalloproteinases (SVMPs), and inclusion of animals from throughout the species' range. For example, *Margres et al. (2015a)* identified only seven SVMP paralogs, and we identified 15. The putative toxins and their abundances are provided in Table 3.

To assess the frequency of presence/absence variation of toxin transcripts in venom-gland transcriptomes for *C. adamanteus* and to determine the potential for missing toxins by sequencing glands from a single individual as the representative of the entire species, we tested for the presence of each of our 59 consensus transcripts in the ten new transcriptomes (Table 4). Seventeen of the 59 transcripts (29%) were absent in at least one of the ten transcriptomes. Only one transcriptome had all 59 (ENP-J), and the most missing was 13 of 59 (CAL-J). The average number missing per transcriptome was 4.5 of 59 (7.6%). The average numbers missing for adults and juveniles were 3.2 and 5.8, respectively, and these numbers were not significantly different (Welch two-sample *t* test: $p = 0.31$). The use of a single representative therefore would likely only result in missing a small number of generally low-expression putative toxins, and adults and juveniles should give equivalent characterizations in terms of the toxins identified, even for a species known to show significant geographic (*Margres et al., 2015a*) and ontogenetic (*Margres et al., 2015b*; *Wray et al., 2015*) expression variation. This result, however, may depend on high sequencing coverage; *Durban et al. (2013)* found more presence/absence differences between a single adult and juvenile venom-gland transcriptome pair for *Crotalus simus*, but their 454 sequencing approach yielded more than an order of magnitude fewer sequenced base pairs per individual.

The number of merged reads per sample (Table 2) ranged from 9,555,490 (CAL-J) to 20,447,473 (ENP-J). To determine whether this variation among samples contributed to the presence/absence patterns described above, we repeated the analysis using 9.5

Rokyta et al. (2017), *PeerJ*, DOI 10.7717/peerj.3249

**Table 3** Transcript abundances of putative toxins comprising the consensus transcriptome for *Crotalus adamanteus*. Values are given in transcripts per million (TPM) as estimated by RSEM.

| Toxin | ANF-A | ANF-J | BR-A | BR-J | CAL-A | CAL-J | ENP-A | ENP-J | LSG-A | LSG-J |
|---|---|---|---|---|---|---|---|---|---|---|
| 3FTx-1 | 4.9 | 34.3 | 0.0 | 0.0 | 0.0 | 0.0 | 31.8 | 72.8 | 208.0 | 0.0 |
| 3FTx-2 | 1.3 | 26.4 | 0.0 | 0.0 | 55.9 | 0.0 | 2.3 | 14.8 | 126.9 | 0.0 |
| BPP-1 | 53,071.6 | 158.1 | 86,213.5 | 187.4 | 128,985.6 | 286.2 | 159,027.1 | 38,210.4 | 13,477.2 | 537.4 |
| CRISP-1 | 4,212.4 | 1,816.6 | 7,813.9 | 1,048.2 | 12,682.6 | 2,585.5 | 5,019.5 | 737.7 | 2,913.9 | 831.4 |
| CTL-1 | 51,386.2 | 63,742.6 | 53,616.6 | 89,233.9 | 95,086.4 | 97,080.1 | 59,762.9 | 112,292.5 | 52,000.6 | 94,736.0 |
| CTL-2 | 12,441.0 | 7,908.6 | 11,010.3 | 20,293.1 | 17,034.3 | 25,689.6 | 19,719.6 | 32,860.3 | 10,156.3 | 22,073.0 |
| CTL-3 | 973.8 | 1,078.3 | 737.2 | 3,454.1 | 1,442.6 | 2,686.4 | 1,753.7 | 2,887.8 | 1,350.7 | 1,658.6 |
| CTL-4 | 5,678.0 | 6,356.9 | 6,869.3 | 10,195.6 | 7,813.3 | 13,458.5 | 9,118.3 | 12,686.1 | 5,080.9 | 10,072.9 |
| CTL-5 | 9,954.2 | 7,948.3 | 10,814.6 | 18,836.6 | 16,923.9 | 29,450.0 | 17,300.5 | 34,528.5 | 10,705.3 | 18,995.7 |
| CTL-6 | 25,421.9 | 25,037.9 | 30,921.3 | 24,830.3 | 48,102.2 | 26,066.8 | 51,036.2 | 45,545.9 | 28,498.6 | 21,708.6 |
| CTL-7 | 964.9 | 2,048.7 | 385.9 | 4,328.4 | 1,273.5 | 2,866.0 | 1,189.7 | 2,679.1 | 1,346.6 | 2,615.7 |
| CTL-8 | 46,187.0 | 52,315.6 | 37,463.5 | 84,319.7 | 87,363.6 | 81,965.5 | 53,409.5 | 93,030.7 | 46,803.0 | 83,239.0 |
| CTL-9 | 2,198.3 | 3,429.7 | 4,724.3 | 6,745.3 | 6,765.7 | 8,127.2 | 3,788.2 | 7,211.4 | 3,195.4 | 5,055.0 |
| CTL-10 | 27,089.7 | 35,138.3 | 36,287.6 | 26,398.0 | 52,737.4 | 22,516.0 | 45,983.0 | 54,230.4 | 33,430.3 | 21,842.3 |
| CTL-11 | 50.7 | 80.9 | 47.1 | 116.7 | 88.3 | 3,465.0 | 38.7 | 140.6 | 45.0 | 150.7 |
| CTL-12 | 0.7 | 4.7 | 0.0 | 0.0 | 0.0 | 0.0 | 7.0 | 26.1 | 37.1 | 0.3 |
| CTL-13 | 13.8 | 85.8 | 0.6 | 6.1 | 2.6 | 315.5 | 10.9 | 26.2 | 11.9 | 12.1 |
| HYAL-1 | 43.3 | 177.6 | 457.7 | 421.9 | 535.3 | 146.5 | 384.4 | 508.5 | 93.5 | 290.7 |
| KUN-1 | 284.9 | 77.3 | 189.9 | 168.0 | 261.6 | 250.1 | 462.4 | 346.3 | 91.8 | 98.7 |
| KUN-2 | 20.8 | 9.8 | 17.6 | 18.0 | 26.6 | 58.5 | 37.6 | 30.6 | 9.3 | 8.8 |
| LAAO-1 | 4,212.0 | 2,024.9 | 12,398.7 | 3,905.8 | 7,273.1 | 217.8 | 14,383.4 | 27,405.7 | 7,619.6 | 4,373.3 |
| MYO-1 | 345,599.5 | 584,312.9 | 362,582.0 | 383,003.8 | 85,226.1 | 445,793.6 | 104,366.4 | 367.4 | 528,354.5 | 491,846.2 |
| MYO-2 | 39,894.7 | 57,390.3 | 120,236.3 | 52,970.6 | 57.5 | 45,995.1 | 45.3 | 64.4 | 62,678.4 | 44,084.5 |
| NGF-1 | 934.6 | 14.9 | 1,303.3 | 17.6 | 1,366.3 | 1.1 | 1,714.4 | 199.9 | 334.1 | 15.5 |
| NUC-1 | 584.3 | 568.8 | 952.9 | 968.4 | 1,334.4 | 1,343.3 | 879.9 | 1,372.4 | 538.0 | 736.3 |
| PDE-1 | 267.6 | 326.4 | 616.0 | 451.3 | 741.1 | 2,394.6 | 442.1 | 989.5 | 157.5 | 353.5 |
| PLA2-1 | 73,635.0 | 50,512.0 | 19,913.5 | 129,007.9 | 23,642.6 | 87,609.4 | 101,745.6 | 168,921.5 | 22,293.7 | 69,631.6 |
| PLA2-2 | 419.7 | 18.1 | 1,070.0 | 175.8 | 2,103.7 | 4.8 | 643.1 | 253.3 | 1,316.8 | 25.0 |
| PLB-1 | 1,169.8 | 200.1 | 1,780.1 | 890.1 | 1,703.4 | 183.2 | 2,057.8 | 1,341.6 | 930.6 | 825.9 |
| SVMPII-1 | 2,935.9 | 16.8 | 894.7 | 26.4 | 8,961.3 | 28.5 | 2,417.1 | 2,287.5 | 4.6 | 12.5 |
| SVMPII-2 | 14,706.7 | 11,144.9 | 9,131.7 | 13,628.9 | 20,869.0 | 18,161.1 | 9,169.5 | 45,877.7 | 8,186.0 | 11,976.5 |
| SVMPII-3 | 8,403.7 | 85.8 | 13,420.9 | 122.9 | 19,674.9 | 96.0 | 3,859.9 | 8,678.2 | 7,624.9 | 96.1 |
| SVMPII-4 | 4,049.2 | 32,884.1 | 33.6 | 26,825.5 | 465.2 | 40,080.3 | 1,069.5 | 65,273.7 | 76.8 | 23,508.1 |
| SVMPII-5 | 133.8 | 1.4 | 130.5 | 31.7 | 243.8 | 52.5 | 274.1 | 245.0 | 58.8 | 8.9 |

Rokyta et al. (2017), PeerJ, DOI 10.7717/peerj.3249

**Table 3** (*continued*)

| Toxin | ANF-A | ANF-J | BR-A | BR-J | CAL-A | CAL-J | ENP-A | ENP-J | LSG-A | LSG-J |
|---|---|---|---|---|---|---|---|---|---|---|
| SVMPIII-1 | 10,470.4 | 20.3 | 4,165.3 | 38.5 | 13,005.4 | 19.6 | 3,558.5 | 3,192.1 | 1,649.2 | 36.0 |
| SVMPIII-2 | 81.8 | 1,020.4 | 10.8 | 1,331.2 | 34.1 | 1,841.7 | 30.7 | 3,354.8 | 2.9 | 717.6 |
| SVMPIII-3 | 1,101.1 | 655.4 | 1,571.5 | 1,644.6 | 3,125.8 | 575.7 | 2,258.2 | 3,107.0 | 897.0 | 1,277.2 |
| SVMPIII-4 | 1,967.6 | 14,665.0 | 3,276.2 | 14,444.0 | 4,069.7 | 8,979.9 | 1,126.3 | 8,554.7 | 1,692.6 | 21,122.7 |
| SVMPIII-5 | 11,432.5 | 14.8 | 37,376.4 | 50.4 | 66,467.1 | 21.6 | 5,538.5 | 21,270.8 | 12,774.1 | 30.7 |
| SVMPIII-6 | 1,049.4 | 5.8 | 1,440.1 | 33.7 | 7,217.5 | 14.0 | 959.9 | 149.0 | 958.6 | 26.8 |
| SVMPIII-7 | 1,589.1 | 39.4 | 2,898.5 | 552.5 | 4,591.6 | 47.6 | 1,483.7 | 3,986.9 | 1,562.0 | 136.6 |
| SVMPIII-8 | 7,083.7 | 7,270.7 | 15,148.7 | 10,705.2 | 23,978.5 | 7,246.1 | 5,059.9 | 26,521.8 | 5,618.1 | 11,705.0 |
| SVMPIII-9 | 410.8 | 698.9 | 69.3 | 135.8 | 164.4 | 771.5 | 142.2 | 162.0 | 53.0 | 156.2 |
| SVMPIII-10 | 1,904.7 | 1,570.5 | 2,132.2 | 3,537.7 | 3,682.4 | 1,198.9 | 1,430.1 | 1,808.5 | 1,822.8 | 3,109.8 |
| SVSP-1 | 3,212.0 | 198.3 | 984.0 | 3,888.3 | 4,673.2 | 1,224.2 | 8,141.1 | 4,747.8 | 1,234.5 | 692.6 |
| SVSP-2 | 59,597.7 | 13,741.0 | 15,149.6 | 21,203.4 | 23,261.7 | 5,467.9 | 24,211.7 | 21,053.0 | 23,635.2 | 15,030.7 |
| SVSP-3 | 28,867.7 | 2,698.0 | 12,881.5 | 8,499.8 | 14,023.4 | 2,218.3 | 14,985.7 | 5,532.7 | 17,129.3 | 2,682.5 |
| SVSP-4 | 4,928.5 | 709.0 | 2,538.4 | 4,983.4 | 11,701.6 | 1,970.9 | 24,651.4 | 18,537.7 | 2,679.1 | 937.2 |
| SVSP-5 | 10,108.4 | 1,331.1 | 5,963.2 | 7,293.7 | 12,403.7 | 1,500.3 | 28,764.4 | 26,790.9 | 2,859.5 | 1,464.5 |
| SVSP-6 | 23,404.3 | 88.6 | 22,095.4 | 4,622.9 | 69,975.9 | 360.9 | 100,840.6 | 60,273.2 | 18,518.5 | 103.2 |
| SVSP-7 | 991.9 | 14.8 | 366.8 | 391.9 | 1,366.8 | 46.9 | 548.2 | 176.6 | 829.0 | 1.9 |
| SVSP-8 | 204.9 | 4.7 | 73.5 | 61.7 | 261.3 | 0.0 | 556.6 | 163.3 | 85.9 | 1.9 |
| SVSP-9 | 15,110.9 | 5.0 | 4,493.4 | 1,668.1 | 52,151.0 | 21.3 | 78,083.6 | 14,284.9 | 4,042.9 | 8.6 |
| SVSP-10 | 72,896.1 | 5,954.9 | 32,324.0 | 7,854.4 | 23,067.6 | 3,146.0 | 10,182.4 | 3,416.3 | 48,775.3 | 8,161.3 |
| SVSP-11 | 2,968.0 | 60.1 | 753.0 | 1,519.2 | 4,619.7 | 119.6 | 8,866.7 | 2,901.5 | 821.3 | 113.4 |
| SVSP-12 | 1,016.0 | 45.9 | 454.1 | 234.7 | 1,074.5 | 19.0 | 2,670.4 | 1,587.9 | 463.4 | 66.2 |
| VEGF-1 | 568.0 | 568.9 | 330.7 | 1,331.7 | 1,383.6 | 2,524.5 | 1,759.7 | 2,668.9 | 376.9 | 279.5 |
| VEGF-2 | 59.2 | 28.5 | 97.6 | 115.6 | 317.6 | 70.6 | 292.6 | 364.4 | 31.8 | 34.9 |
| Vespryn-1 | 2,029.5 | 1,612.2 | 1,371.0 | 1,230.1 | 2,538.4 | 1,618.2 | 2,705.7 | 4,048.9 | 1,730.7 | 682.9 |

**Notes.**

Abbreviations: 3FTx, three-finger toxin; BPP, bradykinin-potentiating and C-type natriuretic peptides; CRISP, cysteine-rich secretory protein; CTL, C-type lectin; HYAL, hyaluronidase; KUN, Kunitz-type protease inhibitor; LAAO, L-amino-acid oxidase, MYO–myotoxin-A; NGF, nerve growth factor; NUC, nucleotidase; PDE, phosphodiesterase; PLA2, phospholipase A$_2$; PLB, phospholipase B; SVMP, snake venom metalloproteinase; SVSP, snake venom serine proteinase; VEGF, vascular endothelial growth factor.

Rokyta et al. (2017), PeerJ, DOI 10.7717/peerj.3249

**Table 4  Putative toxin-encoding transcripts with presence/absence variation.** Transcripts were considered absent if >10% of the coding sequence had < 5× coverage.

| Transcript | All merged reads | | | | | | | | | | 9.5M merged reads | | | | | | | | | |
|---|---|---|---|---|---|---|---|---|---|---|---|---|---|---|---|---|---|---|---|---|
| | ANF | | LSG | | BR | | CAL | | ENP | | ANF | | LSG | | BR | | CAL | | ENP | |
| | A | J | A | J | A | J | A | J | A | J | A | J | A | J | A | J | A | J | A | J |
| 3FTx-1 | − | + | + | − | − | − | − | − | + | + | − | + | + | − | − | − | − | − | + | + |
| 3FTx-2 | − | + | + | − | − | − | + | − | − | + | − | + | + | − | − | − | + | − | − | + |
| CTL-11 | + | + | + | + | + | + | + | + | − | + | + | + | + | + | + | + | + | + | − | + |
| CTL-12 | − | + | + | − | − | − | − | − | − | + | − | − | + | − | − | − | − | − | − | − |
| CTL-13 | + | + | + | + | − | + | − | + | + | + | + | + | + | + | − | + | − | + | − | + |
| NGF-1 | + | + | + | + | + | + | + | − | + | + | + | + | + | + | + | + | + | − | + | + |
| PLA2-2 | + | + | + | + | + | + | + | + | + | + | + | + | + | + | + | + | + | − | + | + |
| SVMPII-1 | + | − | + | − | + | + | + | − | + | + | + | − | + | − | + | − | + | − | + | + |
| SVMPII-3 | + | + | + | + | + | + | + | − | + | + | + | − | + | + | + | + | + | − | + | + |
| SVMPII-5 | + | − | + | − | + | + | + | − | + | + | + | − | + | − | + | + | + | − | + | + |
| SVMPIII-1 | + | − | + | − | + | − | + | − | + | + | + | − | + | − | + | − | + | − | + | + |
| SVMPIII-2 | + | + | − | + | − | + | + | + | + | + | + | + | − | + | − | + | + | + | + | + |
| SVMPIII-5 | + | + | + | + | + | + | + | + | + | + | + | − | + | + | + | + | + | + | + | + |
| SVMPIII-6 | + | − | + | + | + | + | + | − | + | + | + | − | + | + | + | + | + | − | + | + |
| SVMPIII-7 | + | + | + | + | + | + | + | − | + | + | + | + | + | + | + | + | + | − | + | + |
| SVSP-3 | + | + | + | + | + | + | + | + | − | + | + | + | + | + | + | + | + | + | − | − |
| SVSP-7 | + | + | + | − | + | + | + | − | + | + | + | + | + | − | + | + | + | − | + | + |
| SVSP-8 | + | + | + | − | + | + | + | − | + | + | + | + | + | − | + | + | + | − | + | + |
| SVSP-9 | + | + | + | + | + | + | + | + | + | + | + | − | + | + | + | + | + | + | + | + |
| SVSP-12 | + | + | + | + | + | + | + | − | + | + | + | + | + | + | + | + | + | − | + | + |
| # Absent | 3 | 4 | 1 | 8 | 5 | 4 | 3 | 13 | 4 | 0 | 3 | 8 | 1 | 8 | 5 | 5 | 3 | 14 | 5 | 2 |

million merged reads per sample (Table 4). This change increased the number of transcripts showing presence/absence variation from 17 to 20 of 59, and eliminated the one transcriptome (ENP-J) that had all 59 transcripts. A single transcriptome (LSG-A) lacked only a single transcript. The average number missing per transcriptome increased from 4.5 to 5.4. The average missing for adults was 3.4, the average missing for juveniles was 7.4, and these values were not significantly different (Welch two-sample $t$ test: $p = 0.12$). Variation in the number of reads per sample therefore did have minor effects on the detectability of some transcripts in some transcriptomes, but these effects did not change our conclusions above.

Because *C. adamanteus* undergoes a significant ontogenetic change in venom composition (*Margres et al., 2015b*; *Wray et al., 2015*), we looked for evidence that certain toxins were absent from adults or juveniles (Table 4). None of the transcripts were entirely unique to either adults or juveniles, suggesting that the ontogenetic change is more quantitative than qualitative. Of the 17 putative toxin transcripts showing presence/absence variation, six (CTL-11, NGF-1, SVMPII-3, SVMPIII-7, and SVSP-12) were only missing from one of the ten transcriptomes. Four of these six were uniquely missing from the CAL-J transcriptome. Three other transcripts (3FTx-1, 3FTx-2, and CTL-12) showed no particular pattern in terms of ontogeny. Of the remaining eight transcripts, six showed a pattern of being present in all adults, but missing from at least two juveniles. These included two type II metalloproteinases (SVMPII-1 and SVMPII-5), two type III metalloproteinases (SVMPIII-1 and SVMPIII-6), and two serine proteinases (SVSP-7 and SVSP-8). The remaining two of 17, CTL-13 and SVMPIII-2, were present in all juveniles, but absent in two adults. Although this analysis only considered qualitative differences in venom composition, we found some evidence for both adult- and juvenile-associated toxins. Adults appear to generally show a higher complexity of proteases, which may be related to the digestive functions of venoms (*Mackessy, 2010*).

## The repeatability of transcript abundance estimates

Because venom-gland transcriptome sequencing necessitates the sacrifice of animals, we will always be limited in the number of true biological replicates that can be performed, particularly for vertebrate species like *C. adamanteus* that are of conservation concern and typically have relatively small population sizes. To maximize the information gained for each individual snake and to assess the repeatability of our estimates of transcript abundances, we sequenced the left and right venom glands of each individual separately. We found nearly perfect agreement between toxin-transcript abundance estimates across glands for each of our ten individuals (Fig. 2). We measured abundances in transcripts per million (TPM) using RSEM (*Li & Dewey, 2011*) and used a centered logratio (clr) transform (*Aitchison, 1986*). This monotonic transform has no effect on rank-order relationships and is equivalent to a log transform for linear relationships. Spearman's rank correlation coefficients ($\rho$) ranged from 0.97 through 1.0 (Fig. 2), with six of ten individuals showing exact agreement (i.e., $\rho = 1.0$). Pearson's correlation coefficients range from 0.90 through 1.0 (Fig. 2), with four of ten showing essentially perfect linear correspondence (i.e., $R = 1.0$). Our transcript abundance estimates for toxin transcripts were therefore highly repeatable,

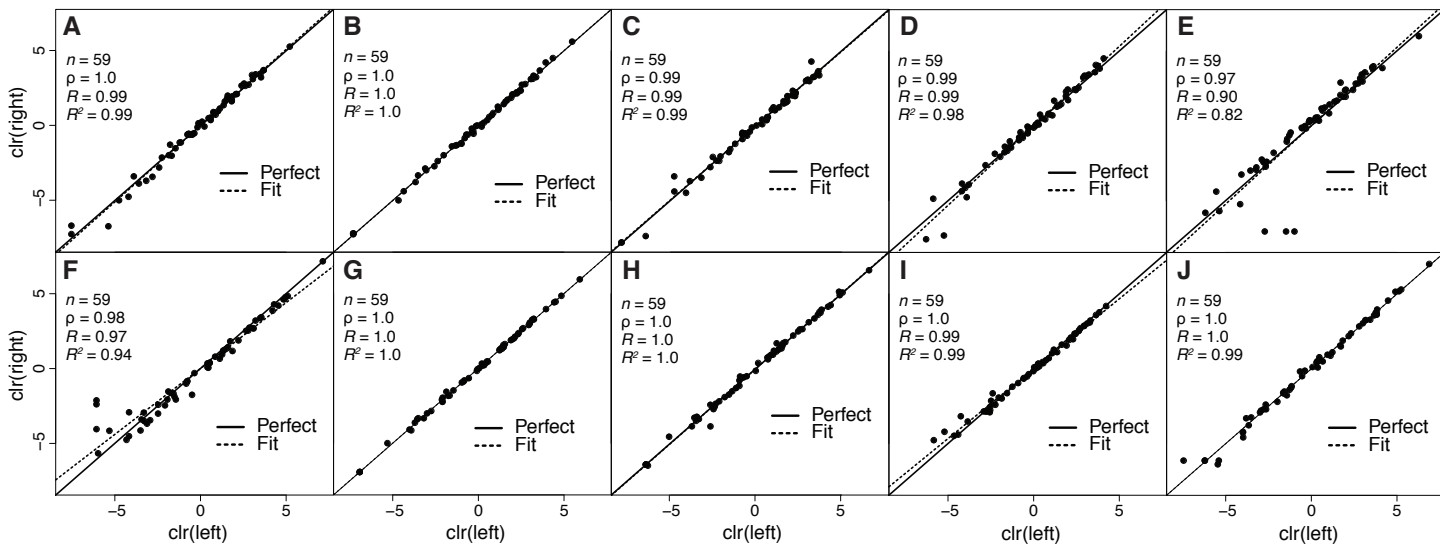

**Figure 2 Estimates of toxin expression levels match nearly exactly across left and right glands of individual snakes.** The solid diagonal line corresponds to perfect agreement. All rank correlations were ≥0.97, and all linear correlation coefficients were ≥0.90, with nine of ten ≥0.97. (A) Apalachicola National Forest adult (ANF-A, KW1264). (B) Brooksville Ridge adult (BR-A, MM0127). (C) Caladesi Island adult (CAL-A, KW1942). (D) Everglades National Park adult (ENP-A, KW0944). (E) Little St. George Island adult (LSG-A, KW2161). (F) Apalachicola National Forest juvenile (ANF-J, MM0198). (G) Brooksville Ridge juvenile (BR-J, KW2171). (H) Caladesi Island juvenile (CAL-J, KW2170). (I) Everglades National Park juvenile (ENP-J, MM0143). (J) Little St. George Island juvenile (LSG-J, KW2184). Abbreviations: clr, centered logratio; $n$, number of transcripts; $\rho$, Spearman's rank correlation coefficient; $R$, Pearson's correlation coefficient; $R^2$, coefficient of determination.

and we found no evidence for different expression patterns between left and right glands for individuals. This analysis demonstrated that, for statistical purposes, left and right glands are similar to technical replicates. For characterizing venom composition of a single individual, however, we found no clear benefit to separately sequencing the two glands.

## Ontogenetic expression variation

The presence of an ontogenetic venom change in *C. adamanteus* is well documented. Venom proteomic composition changes through the first postnatal shed (*Wray et al., 2015*) and appears to have a defined shift at sexual maturity (*Margres et al., 2015b*). *Margres et al. (2015b)* used a reversed-phase high performance liquid chromatography (RP-HPLC) approach to quantify ontogenetic and geographic venom expression differentiation for 25 protein peaks and found that the majority of the ontogenetic change could be attributed to five of 25 peaks. They identified proteins encoded by nine loci from a juvenile venom-gland transcriptome for *C. adamanteus* (*Rokyta et al., 2012*) in these five RP-HPLC fractions, but four of the five peaks had three or more detectable proteins, so the specific loci involved in the ontogenetic change could not be unambiguously identified. The candidate loci they identified included C-type lectins (CTLs), L-amino acid oxidase (LAAO), type II and type II snake-venom metalloproteinases (SVMPII and SVMPIII), and serine proteinases (SVSPs). This list of loci probably includes loci not involved in the ontogenetic change but that happen to elute in the same peaks as loci that are involved in the change, and the list is possibly incomplete because of the reliance on a single juvenile transcriptome for the

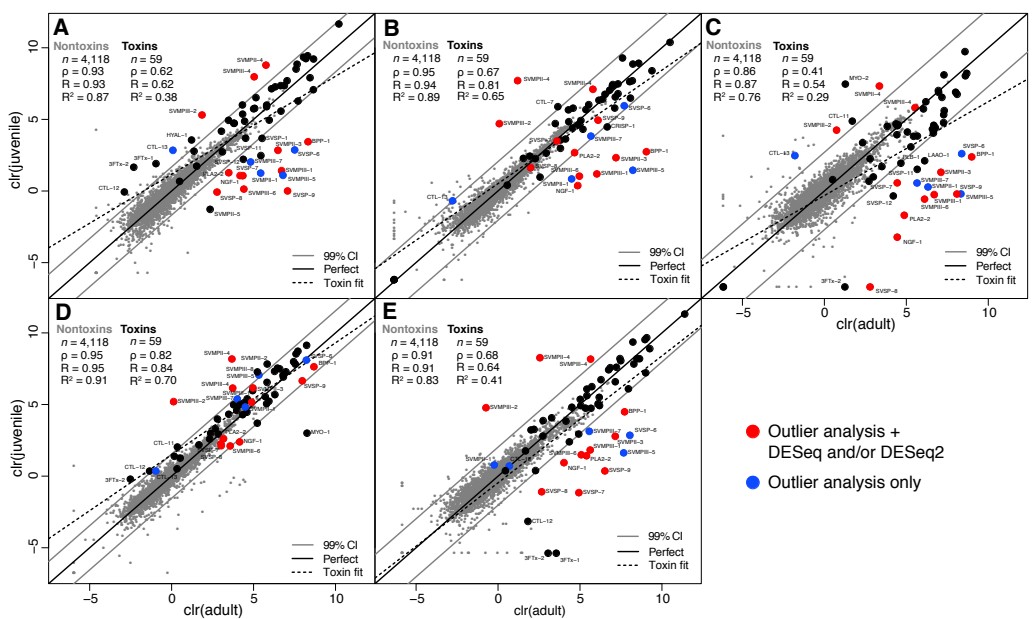

**Figure 3** **Adult/juvenile expression comparisons identified 17 differentially expressed toxin transcripts.** Nontoxin expression comparisons were used to establish a null distribution and 99% confidence interval for differences for each of five comparisons. Transcripts indicated by blue dots were those detected as outliers with expression differences in the same direction in at least three of five comparisons. Transcripts indicated with red dots were also detected as differentially expressed by DESeq and/or DESeq2 (Table 5). (A) Apalachicola National Forest (ANF). (B) Brooksville Ridge (BR). (C) Caladesi Island (CAL). (D) Everglades National Park (ENP). (E) Little St. George Island (LSG). Abbreviations: clr, centered logratio; *n*, number of transcripts; $\rho$, Spearman's rank correlation coefficient; *R*, Pearson's correlation coefficient; $R^2$, coefficient of determination.

identification of proteins in each peak. As shown above, any single individual transcriptome is likely to be missing loci for the species.

We collected adult/juvenile pairs of *C. adamanteus* from the five populations shown by *Margres et al. (2015a)* and *Margres et al. (2015b)* to have different venom compositions (Fig. 1). For each of our five adult/juvenile venom-gland transcriptome pairs, we identified toxin transcripts outside a 99% confidence interval estimate on the basis of nontoxin transcripts (Fig. 3). Toxins outside these confidence intervals in a majority of comparisons (i.e., three of five) with expression changes in the same direction for a majority of comparisons (e.g., upregulated in adults relative to juveniles) were considered to be involved in the ontogenetic change. This outlier approach, requiring a majority consensus, ensured that interpopulation expression variation not related to ontogeny would not result in false positives. Of 59 total toxin transcripts, we identified 13 toxin loci that were upregulated in adults relative to juveniles and four loci that were downregulated in adults relative to juveniles (Table 5). The juvenile-biased toxins included a C-type lectin (CTL), a type II snake-venom metalloproteinase (SVMPII), and two type III snake-venom metallproteinases (SVMPIIIs). The adult-biased toxins included Bradykinin-potentiating and C-type natriuretic peptides (BPP), nerve growth factor (NGF), a phospholipase A₂ (PLA2), two SVMPIIs, four SVMPIIIs, and four snake-venom serine proteinases (SVSPs).

**Table 5 Ontogenetic differential expression analyses for *Crotalus adamanteus*.** Highlighted rows represent the consensus candidates of ontogenetically variable loci. The "NA" values from DESeq2 result from expression levels of zero.

| Toxin | Outlier analysis (A relative to J) | | | | | | DESeq | | DESeq2 | |
|---|---|---|---|---|---|---|---|---|---|---|
| | ANF | BR | CAL | ENP | LSG | $\Delta_{J \to A}$ | $P_{adj}$ | $\log_2 \Delta_{J/A}$ | $P_{adj}$ | $\log_2 \Delta_{J/A}$ |
| 3FTx-1 | ↓ | – | – | – | ↑ | – | > 0.1 | −1.82 | NA | −0.08 |
| 3FTx-2 | ↓ | – | ↑ | ↓ | ↑ | – | > 0.1 | −2.09 | NA | −0.12 |
| BPP-1 | ↑ | ↑ | ↑ | – | ↑ | Up | $1.8 \times 10^{-3}$ | −3.55 | $8.3 \times 10^{-2}$ | −0.83 |
| CRISP-1 | – | ↑ | – | – | – | – | > 0.1 | −1.85 | $2.7 \times 10^{-2}$ | −1.07 |
| CTL-1 | – | – | – | – | – | – | > 0.1 | 0.77 | > 0.1 | 0.55 |
| CTL-2 | – | – | – | – | – | – | > 0.1 | 0.81 | > 0.1 | 0.63 |
| CTL-3 | – | – | – | – | – | – | > 0.1 | 0.97 | > 0.1 | 0.71 |
| CTL-4 | – | – | – | – | – | – | > 0.1 | 0.81 | > 0.1 | 0.64 |
| CTL-5 | – | – | – | – | – | – | > 0.1 | 0.83 | > 0.1 | 0.66 |
| CTL-6 | – | – | – | – | – | – | > 0.1 | −0.02 | > 0.1 | −0.02 |
| CTL-7 | – | ↓ | – | – | – | – | > 0.1 | 1.67 | $2.9 \times 10^{-2}$ | 1.04 |
| CTL-8 | – | – | – | – | – | – | > 0.1 | 0.77 | > 0.1 | 0.53 |
| CTL-9 | – | – | – | – | – | – | > 0.1 | 0.68 | > 0.1 | 0.49 |
| CTL-10 | – | – | – | – | – | – | > 0.1 | 0.05 | > 0.1 | 0.03 |
| CTL-11 | – | – | ↓ | ↓ | – | – | > 0.1 | 1.45 | $2.7 \times 10^{-2}$ | 1.01 |
| CTL-12 | ↓ | – | – | ↓ | ↑ | – | > 0.1 | −1.14 | > 0.1 | −0.18 |
| CTL-13 | ↓ | ↓ | ↓ | – | – | Down | > 0.1 | 3.08 | > 0.1 | 0.65 |
| HYAL-1 | ↓ | – | – | – | – | – | > 0.1 | 0.43 | > 0.1 | 0.23 |
| KUN-1 | – | – | – | – | – | – | > 0.1 | −0.28 | > 0.1 | −0.26 |
| KUN-2 | – | – | – | – | – | – | > 0.1 | 0.15 | > 0.1 | 0.15 |
| LAAO-1 | – | – | ↑ | – | – | – | > 0.1 | −0.37 | > 0.1 | −0.12 |
| MYO-1 | – | – | – | ↑ | – | – | > 0.1 | 0.52 | > 0.1 | 0.18 |
| MYO-2 | – | – | ↓ | – | – | – | > 0.1 | −0.15 | > 0.1 | −0.02 |
| NGF-1 | ↑ | ↑ | ↑ | ↑ | ↑ | Up | $9.3 \times 10^{-13}$ | −4.54 | $2.0 \times 10^{-2}$ | −1.05 |
| NUC-1 | – | – | – | – | – | – | > 0.1 | 0.34 | > 0.1 | 0.26 |
| PDE-1 | – | – | – | – | – | – | > 0.1 | 0.85 | > 0.1 | 0.64 |
| PLA2-1 | – | – | – | – | – | – | $9.4 \times 10^{-2}$ | 1.39 | $2.8 \times 10^{-2}$ | 0.98 |
| PLA2-2 | ↑ | – | ↑ | – | ↑ | Up | $7.1 \times 10^{-2}$ | −3.87 | $1.3 \times 10^{-2}$ | −1.14 |
| PLB-1 | – | – | ↑ | – | – | – | > 0.1 | −0.87 | > 0.1 | −0.51 |
| SVMPII-1 | ↑ | ↑ | ↑ | – | – | Up | > 0.1 | −3.02 | > 0.1 | −0.52 |
| SVMPII-2 | – | – | – | ↓ | – | – | > 0.1 | 0.69 | > 0.1 | 0.48 |
| SVMPII-3 | ↑ | ↑ | ↑ | – | ↑ | Up | $9.4 \times 10^{-2}$ | −2.75 | > 0.1 | −0.75 |
| SVMPII-4 | ↓ | ↓ | ↓ | ↓ | ↓ | Down | $7.8 \times 10^{-7}$ | 5.33 | $3.2 \times 10^{-2}$ | 0.94 |
| SVMPII-5 | ↑ | – | – | – | – | – | > 0.1 | −2.10 | > 0.1 | −0.68 |
| SVMPIII-1 | ↑ | ↑ | ↑ | – | ↑ | Up | $9.4 \times 10^{-2}$ | −3.23 | > 0.1 | −0.68 |
| SVMPIII-2 | ↓ | ↓ | ↓ | ↓ | ↓ | Down | $9.7 \times 10^{-6}$ | 2.65 | $2.3 \times 10^{-8}$ | 1.90 |
| SVMPIII-3 | – | – | – | – | – | – | > 0.1 | 0.11 | > 0.1 | 0.08 |
| SVMPIII-4 | ↓ | – | – | ↓ | ↓ | Down | > 0.1 | 2.96 | $7.9 \times 10^{-4}$ | 1.46 |
| SVMPIII-5 | ↑ | ↑ | ↑ | ↓ | ↑ | Up | > 0.1 | −2.68 | > 0.1 | −0.58 |

Table 5 (*continued*)

| Toxin | Outlier analysis (A relative to J) | | | | | | DESeq | | DESeq2 | |
|-------|------|------|------|------|------|-------------------|-------|-------------------|-------|-------------------|
| | ANF | BR | CAL | ENP | LSG | $\Delta_{J \to A}$ | $P_{adj}$ | $\log_2 \Delta_{J/A}$ | $P_{adj}$ | $\log_2 \Delta_{J/A}$ |
| SVMPIII-6 | ↑ | ↑ | ↑ | – | ↑ | Up | $9.4 \times 10^{-2}$ | −4.70 | $8.3 \times 10^{-4}$ | −1.40 |
| SVMPIII-7 | ↑ | – | ↑ | – | ↑ | Up | > 0.1 | −1.19 | > 0.1 | −0.64 |
| SVMPIII-8 | – | – | – | ↓ | – | – | > 0.1 | 0.36 | > 0.1 | 0.21 |
| SVMPIII-9 | – | – | – | – | – | – | > 0.1 | 1.39 | $9.8 \times 10^{-2}$ | 0.87 |
| SVMPIII-10 | – | – | – | – | – | – | > 0.1 | 0.42 | > 0.1 | 0.23 |
| SVSP-1 | ↑ | – | – | – | – | – | > 0.1 | −0.59 | > 0.1 | −0.34 |
| SVSP-2 | – | – | – | – | – | – | > 0.1 | −0.61 | > 0.1 | −0.33 |
| SVSP-3 | – | – | – | – | – | – | > 0.1 | −1.75 | $2.5 \times 10^{-2}$ | −1.08 |
| SVSP-4 | – | – | – | – | – | – | > 0.1 | −0.60 | > 0.1 | −0.32 |
| SVSP-5 | – | – | – | – | – | – | > 0.1 | −0.53 | > 0.1 | −0.28 |
| SVSP-6 | ↑ | – | ↑ | – | ↑ | Up | > 0.1 | −2.00 | > 0.1 | −0.44 |
| SVSP-7 | ↑ | – | ↑ | – | ↑ | Up | > 0.1 | −1.50 | $5.1 \times 10^{-2}$ | −0.96 |
| SVSP-8 | ↑ | – | ↑ | – | ↑ | Up | $4.8 \times 10^{-2}$ | −1.32 | > 0.1 | −0.81 |
| SVSP-9 | ↑ | – | ↑ | – | ↑ | Up | $2.3 \times 10^{-2}$ | −3.40 | > 0.1 | −0.45 |
| SVSP-10 | – | – | – | – | – | – | > 0.1 | −2.39 | $3.7 \times 10^{-2}$ | −1.04 |
| SVSP-11 | ↑ | – | ↑ | – | – | – | > 0.1 | −1.72 | > 0.1 | −0.54 |
| SVSP-12 | – | – | ↑ | – | – | – | > 0.1 | −2.01 | > 0.1 | −0.45 |
| VEGF-1 | – | – | – | – | – | – | > 0.1 | 0.79 | $2.6 \times 10^{-2}$ | 0.70 |
| VEGF-2 | – | – | – | – | – | – | > 0.1 | −0.23 | > 0.1 | −0.14 |
| Vespryn-1 | – | – | – | – | – | – | > 0.1 | −0.07 | > 0.1 | −0.05 |

**Notes.**

Abbreviations: 3FTx, three-finger toxin; A, adult; BPP, bradykinin-potentiating and C-type natriuretic peptides; CRISP, cysteine-rich secretory protein; CTL, C-type lectin; HYAL, hyaluronidase; J, juvenile; KUN, Kunitz-type protease inhibitor; LAAO, L-amino-acid oxidase; MYO, myotoxin-A; NGF, nerve growth factor; NUC, nucleotidase; PDE, phosphodiesterase; PLA2, phospholipase A$_2$; PLB, phospholipase B; SVMP, snake venom metalloproteinase; SVSP, snake venom serine proteinase; VEGF, vascular endothelial growth factor.

As further confirmation for the results of our outlier analysis, we applied DESeq (*Anders & Huber, 2010*) and DESeq2 (*Love, Huber & Anders, 2014*) to compare our five adult transcriptomes to our five juvenile transcriptomes (Table 5). DESeq identified 35 of 4,177 total transcripts as being differentially expressed with a false-discovery rate (FDR) of 0.1. Of these 35, 24 were nontoxins, and 11 were toxins. DESeq2 identified 182 of 4,177 total transcripts as being differentially expressed between adults and juveniles with an FDR of 0.1, including 166 nontoxins and 16 toxins. The three methods showed good agreement (Table 5). Of the 11 toxins detected by DESeq, only one (PLA2-1) was not detected in the outlier analysis. Of the 16 toxins detected by DESeq2, seven were not detected by the outlier analysis. Three of these seven were outliers in at least one of the five adult/juvenile comparisons but did not meet the stringent (i.e., conservative) criterion of being outliers in a majority of comparisons. Of the 17 toxins detected in the outlier analysis, 12 were confirmed by either DESeq or DESeq2 (Table 5).

The 12 loci detected in the outlier analysis and DESeq and/or DESeq2 represented our best conservative estimate of the loci involved in the ontogenetic change in *C. adamanteus* (highlighted rows in Table 5). Juveniles expressed SVMPII-4, SVMPIII-2, and SVMPIII-4 at higher levels than adults; all of the juvenile-biased toxins were SVMP paralogs. Relative to

juveniles, adults upregulated BPP-1, NGF-1, PLA2-2, SVMPII-3, SVMPIII-1, SVMPIII-6, SVSP-7, SVSP-8, and SVSP-9. Therefore, the primary, range-wide ontogenetic change for *C. adamanteus* involved down regulation of three SVMPs and upregulation of a diverse array of nine toxins from six toxin classes. Snake-venom metalloproteinases induce the local and systemic hemorrhage characteristic of viperid bites and are classified on the basis of their domain structures (*Fox & Serrano, 2005*; *Fox & Serrano, 2010*). All SVMPs have a metalloproteinase domain with a zinc-binding motif. Type II SVMPs (SVMPIIs) have an additional disintegrin domain, which may be proteolytically cleaved posttranslationally to produce a free disintegrin. Type III SVMPs (SVMPIIIs) have additional disintegrin-like and cysteine-rich domains. Adults and juveniles of *C. adamanteus* express similar total amounts of SVMPs. The average for adults was 65,173.7 TPM, and the average for juveniles was 75,654.6 TPM. These averages were not statistically different (Welch two-sample $t$ test: $p = 0.68$). The ontogenetic change therefore resulted in a maintenance of overall SVMP expression with a shift in expression from juvenile to adult paralogs. The adult-biased toxins include a diverse set of toxin types. Snake-venom serine proteinases (SVSPs) interfere with blood coagulation and hemostasis and belong to the trypsin family of serine proteases (*Serrano & Maroun, 2005*; *Phillips, Swenson & Francis S. Markland, 2010*). Bradykinin-potentiating and C-type natriuretic peptides are presumed to cause a reduction in blood pressure (*Pahari, Mackessy & Kini, 2007*). Despite being present in diverse snake venoms, nerve growth factor (NGF) has an unknown role as a component of venoms (*Kostiza & Meier, 1996*; *Lavin et al., 2010*) but was also more highly expressed in the venom-gland transcriptome of an adult *C. simus* than a juvenile (*Durban et al., 2013*). Phospholipases $A_2$ are among the most functionally diverse classes of snake-venom toxins and have pharmocological effects that include neurotoxicity (presynaptic or postsynaptic), myotoxicity, and cardiotoxicity. Anticoagulant and hemolytic effects due to PLA2s are also known (*Lynch, 2007*; *Doley, Zhou & Kini, 2010*). *Crotalus adamanteus* expresses only one PLA2 transcript at a high level, but a second (PLA2-2) is expressed at low levels (Table 3; *Dowell et al., 2016*). The low-expression paralog was detected as a component of the ontogenetic change. Although all of the adult-biased toxins were also expressed in juveniles, their upregulation in adults should result in a more functionally complex venom. Adults and juveniles share the same genomes and are therefore confined to the same sets of available toxin genes. The presence of toxins differentially expressed in adults and juveniles may ultimately allow us to assess the relative strength of selection acting on venoms of adults and juveniles by comparing evolutionary patterns in the sequences of juvenile-biased and adult-biased toxin genes.

   *Durban et al. (2013)* conducted a proteomic and transcriptomic characterization of the ontogenetic change in *C. simus* and showed that the proteome-based venom profiles showed less agreement than the transcriptome-based profiles. In particular, they claimed that the venom-gland transcriptomes were largely indistinguishable between adults and juveniles, despite major divergence in the venom proteomes. They therefore attributed at least some portion of the ontogenetic change to posttranscriptional regulation mediated by microRNAs. They used a single adult and juvenile for transcriptomics and pooled adult and juvenile venoms for proteomics, which leaves no real possiblity of a true statistical

comparison between adults and juveniles or between transcriptomes and proteomes. Conclusions as significant as the implication of specific posttranscriptional regulatory mechanisms are best not made on the basis of samples sizes of one. We showed that the ontogenetic change in *C. adamanteus* has a simple, statistically significant transcriptional basis, which was consistent with the proteomic patterns underlying the ontogenetic change in *C. adamanteus* described by *Margres et al. (2015b)* and the functional characterizations described by *Margres et al. (2016a)*.

## Ontogenetic toxicity variation

To determine whether the significantly different venom profiles for adults and juveniles of *C. adamanteus* detected by *Margres et al. (2015b)*, which are underlain by the transcriptomic differences described above, resulted in different effects in model prey, we measured median lethal doses ($LD_{50}$s) of pooled adult and juvenile venoms in mice. We used venoms from adults and juveniles from the Apalachicola National Forest (ANF) population, because the venom-gland transcriptomic comparison for this population (Fig. 3) showed all of the differences detected in our outlier analysis. We estimated an adult $LD_{50}$ of 3.46 mg/kg and a juvenile $LD_{50}$ of 2.79 mg/kg, indicating that the juvenile venom was more potent in mice. The differential venom-transcript expression described above therefore translates to a toxicity difference in the venoms. Although this toxicity difference was detected in a model prey species, the difference clearly indicated large phenotypic differences in the functions and actions of adult and juvenile venoms. The difference therefore implies a difference in the effects of these venom in natural prey species, although the magnitude and direction of the difference is likely to be different in natural prey. In other species with known ontogenetic venom compositional changes, juveniles tend to show more toxic venoms (*Mackessy, 1988*; *Saldarriaga et al., 2003*; *Mackessy et al., 2006*). This result may depend, however, on the choice of prey species used in the toxicity assays. Juveniles of *Bothrops jararaca* have more toxic venoms than adults in chicks, but adults are more potent against mice (*Zelanis et al., 2010*), suggesting that the difference between adult and juvenile venoms may simply maintain higher toxicity to the different preferred prey of each of the two age classes.

## CONCLUSIONS

Pseudobiological replicates consisting of separate preparation of libraries from left and right glands gave indistinguishable results. Such replicates were therefore equivalent to technical replicates and provide little potential for improving our understanding of venom composition for any particular animal. They did, however, clearly demonstrate the repeatability of our RNA-seq approach. Even with a pronounced ontogenetic change, individual adult and juvenile venom-gland transcriptomes were similarly effective in giving near-complete characterizations of the identities of the full complement of venom genes, despite major differences in transcriptional levels. A single individual (adult or juvenile) will give the vast majority of toxin sequences for the species, but a complete characterization of the venom genes for a species will require more than a single venom-gland transcriptome because of presence/absence differences among transcriptomes. We identified a set of

juvenile toxins that were expressed more highly in juveniles than adults across the range of *C. adamanteus*. These juvenile toxins included one of five SVMPII paralogs and two of 10 SVMPIII paralogs. We identified a set of adult toxins that were expressed at higher levels in adults relative to juveniles. These included BPP, NGF, one of two PLA2 paralogs, one of five SVMPII paralogs, two of 10 SVMPIII paralogs, and three of 12 SVSP paralogs. Comparing patterns of sequence variation in these two ontogenetic classes of toxins could allow us to ascertain the relative strength of selection acting on different life stages of *C. adamanteus* and the relative extent of coevolutionary interactions for adult and juvenile snakes. Adult and juvenile venoms had measurably different effects in mice, suggesting that the detected and characterized ontogenetic change significantly affects venom efficacy in natural prey.

## ACKNOWLEDGEMENTS

We thank Kenneth P. Wray for assistance in acquiring and processing animals. We thank Megan Lamb, Jennifer Wanat, Ethan Bourque, and Rebecca Bernard with the Florida DEP and Apalachicola River NERR and Peter Krulder, Carl Calhoun, and Rick Coosey at Caladesi Island State Park for access to field sites.

### Funding

Funding for this work was provided by the National Science Foundation (DEB 1145987) to DRR and the NCRR/BMRG Viper Resource Grant (P40OD010960-13) to EES. The funders had no role in study design, data collection and analysis, decision to publish, or preparation of the manuscript.

### Grant Disclosures

The following grant information was disclosed by the authors:
National Science Foundation: DEB 1145987.
NCRR/BMRG Viper Resource: P40OD010960-13.

### Competing Interests

The authors declare there are no competing interests.

### Author Contributions

- Darin R. Rokyta conceived and designed the experiments, analyzed the data, wrote the paper, prepared figures and/or tables, reviewed drafts of the paper.
- Mark J. Margres conceived and designed the experiments, performed the experiments, analyzed the data, reviewed drafts of the paper.
- Micaiah J. Ward and Elda E. Sanchez performed the experiments, reviewed drafts of the paper.

## Animal Ethics

The following information was supplied relating to ethical approvals (i.e., approving body and any reference numbers):

The above procedures were approved by the Florida State University Institutional Animal Care and Use Committee (IACUC) under protocol #0924.

## Field Study Permissions

The following information was supplied relating to field study approvals (i.e., approving body and any reference numbers):

Specimens were collected under the following permits: Florida Fish and Wildlife Conservation Commission (FWC) LSSC-13-00004 and LSSC-09-0399 and Florida Department of Environmental Protection permit #03131424.

## DNA Deposition

The following information was supplied regarding the deposition of DNA sequences:

The raw transcriptome reads were submitted to the National Center for Biotechnology Information (NCBI) Sequence Read Archive (SRA) under BioProject PRJNA88989, BioSamples SAMN06338134(KW0944-L), SAMN06338135 (KW0944-R), SAMN06338136 (KW1264-L), SAMN06338137(KW1264-R), SAMN06338138 (KW1942-L), SAMN06338139 (KW1942-R), SAMN06338140 (KW2161-L), SAMN06338141 (KW2161-R), SAMN06338142 (KW2170-L), SAMN06338143 (KW2170-R), SAMN06338144 (KW2171-L), SAMN06338145 (KW2171-R), SAMN06338146 (KW2184-L), SAMN06338147 (KW2184-R), SAMN06338148 (MM0127-L), SAMN06338149 (MM0127-R), SAMN06338150 (MM0143-L), SAMN06338151 (MM0143-R), SAMN06338152 (MM0198-L), SAMN06338153 (MM0198-R), and SRA accessions SRR5259500 (KW0944-L), SRR5259499 (KW0944-R), SRR5259498 (KW1264-L), SRR5259497 (KW1264-R), SRR5259496 (KW1942-L), SRR5259495 (KW1942-R), SRR5259494 (KW2161-L), SRR5259493 (KW2161-R), SRR5259492 (KW2170-L), SRR5259491 (KW2170-R), SRR5259490 (KW2171-L), SRR5259489 (KW2171-R), SRR5259488 (KW2184-L), SRR5259487 (KW2184-R), SRR5259486 (MM0127-L), SRR5259485 (MM0127-R), SRR5259484 (MM0143-L), SRR5259483 (MM0143-R), SRR5259482 (MM0198-L), SRR5259481 (MM0198-R).

The abbreviation "L" refers to the left venom gland, and "R" refers to the right venom gland. The consensus transcripts were submitted to the NCBI Transcriptome Shotgun Assembly database. This Transcriptome Shotgun Assembly project has been deposited at DDBJ/EMBL/GenBank under the accession GFHW00000000. The version described in this paper is the first version, GFHW01000000.

## Data Availability

The raw transcriptome reads were submitted to the National Center for Biotechnology Information (NCBI) Sequence Read Archive (SRA).

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
