# Peer review of "The genetics of venom ontogeny in the eastern diamondback rattlesnake (Crotalus adamanteus)"

_PeerJ, doi:10.7717/peerj.3249_

## Round 0.1 · original submission · Major Revisions

· Academic Editor

Major Revisions

Two reviewers have now reviewed the manuscript, and both find merit in it. Reviewer 1 asks for clarification about discussion of previous literature, and Reviewer 2 brings up a number of questions about methodological points. The reviewer comments seem reasonable, and while there is no need for example to redo de novo assembly using other tools, clarification of the methods and discussion of the impact of different approaches would be welcome.

·

Basic reporting

no comment

Experimental design

no comment

Validity of the findings

no comments

Additional comments

To characterize the genetics underlying the ontogenetic venom compositional change in C. adamanteus, Rokyta and co-workers sequenced adult/juvenile pairs of venom-gland transcriptomes from five populations previously shown to have different adult venom compositions. From a total of 59 putative toxin transcripts identified for C. adamanteus, were involved in the ontogenetic change. The work adds valuable information to our understanding of the molecular ontogenetic changes occuring in C. adamanteus venom. However, from there to postulate that "the same selective forces that give rise to rapid inter- and intraspecific divergence in snake venoms can also favor differences in venoms across life-history stages", much research remains to be done! Also, in some passages the authors are aggressively critisize previous work conducted by other research groups, in particular by Durban et al. Not only does this attitude seem unjustified, but they themselves fall into the same thing they criticize. I would recommend the work of Rokyta et al for publication in Peer J. once it has been revised according to the following points:

".... has one of the most well-characterized venom-gland transcriptomes (Rokyta et al., 2011, 2012, 2015a) and venom proteomes of any snake species (Margres et al., 2014)". The work of Eichberg and colleagues deserves (Expert Rev Proteomics 2015;12:557-73) to be cited as well.

" Although many studies have demonstrated the presence of an ontogenetic change in venom composition and have even identified the types of toxins involved, none have resolved the nature of the change to particular paralogs within the context of a complete species-level characterization of venom composition". Please consult Durban et al. BMC Genomics 2013, 14:234.

The authors critisize Durban et al for using "a single adult and juvenile for transcriptomics and pooled adult and juvenile venoms for proteomics, which leaves no real possiblity of a true statistical comparison between adults and juveniles or between transcriptomes and proteomes". However, what Rokyta and co-workers do not say is that the study of Durban et al. included the temporal monitoring of the ontogenetic changes of venoms from a large number (>25) of adult Crotalus simus specimens from 11 captive-born siblings from an age of 8-weeks to 21-months. This age-resolved analysis showed that ontogenetic changes are qualitatively indistinguishable between individuals, although it may quantitatively vary in the age at which they occur in different individuals. In addition, the difference between neonates and adults is so obvious to the naked eye that statistical analysis is required to define the temporal pattern of the ontogenetic shifts.

On the other hand, toxicity assays were conducted with pooled venoms: "To determine whether the significantly different venom profiles for adults and juveniles of C. adamanteus detected by Margres et al. (2015b), which are underlain by the transcriptomic differences described above, resulted in different effects in model prey, we measured median lethal doses (LD50s) of POOLED adult and juvenile venoms in mice", breaking thus individual genotype-phenotype connections...

Reviewer 2 ·

Basic reporting

Rokyta et al. used eastern diamondback rattlesnake to study ontogenetic changes in venom at the transcriptome level. Overall the manuscript is well written but some part is ambiguous in writing. Authors did a nice job to give the background information about venom study in snakes and especially in eastern diamondback rattlesnake and provide information for knowledge gap as well. Author already submitted all the data to the NCBI for data availability. I have some comments below:

1. Different from previous study, author sampled adult/juvenile pair in five populations, and did RNA-Seq sequencing on both left and right venom glands. Based on Figure2, left and right venom glands shows high correlation in toxin expression levels, so authors argued left and right glands as the technical replicate. However, it has been show earlier, the high correlation cannot be used to measure agreement. See paper McIntyre et al. 2011 BMC Genomics (http://bmcgenomics.biomedcentral.com/articles/10.1186/1471-2164-12-293). Also, in Table 2, the amount of data in left and right venom glands varies a lot in each adult or juvenile samples, I would like to see more evidence suggest left and right glands can be used as technical replicate.

2. Table 2, the amount of RNA-Seq data between adult and juvenile samples varies a lot as well. Does this un-equal amount of data would affect the results? For example, if you look at the Table 4 the presence/absence (PAV) variation of putative toxin-coding transcripts. It looks like the number of absent transcripts are related to the amount of data in Table 2, in LSG, CAL and ENP populations. Could author do subsampling to equal amount of data and see whether these PAV transcripts still exist?

3. Line 67: “Although many studies have demonstrated the presence of an ontogenetic change in venom composition and have even identified the types of toxins involved, none have resolved the nature of the change to particular paralogs within the context of a complete species-level characterization of venom composition”. Since the manuscript used the denovo approach to assemble transcripts without mapping to the reference genome, I wonder how authors would domesticated the accurate of the results draw from mapping to the paralogs, how author avoid the potential error caused from estimating the paralogs transcripts abundance?

Experimental design

The research question in this manuscript is well defined and the results from this study is meaning to the snake venom community and fill the knowledge gap about ontogenetic change in transcriptome analysis based on sampling in multiple populations and both in left and right glands. But I have some comments about methods used in this manuscript.

1. Line 126: “assembled with SeqMan NGen version 12.3.1”. Denovo transcriptome assembly without reference genome is a common task now for many studies, why author used the SeqMan NGen to assemble the transcriptome instead of using more popular tools such as Trinity, which suggested give a good result to assemble full length transcripts (http://www.nature.com/nbt/journal/v29/n7/full/nbt.1883.html)?

2. Line 128: “In addition, 1,000 merged reads for each128 individual were used as seeds with our in-house assembler Extender (Rokyta et al., 2012)” So here Extender is another denovo assembly tool, right? If yes, need to be clear in the methods. Another don’t why choose these two denovo transcriptome assembly tools, maybe authors have done some benchmark earlier about these tools comparing to the general tools used, if yes, please cited those works as well.

3. From the methods section, it is not clear to me how author define “paralogs” in this manuscript and how to identify paralogs since these are the major finding difference from previous studies?

4. Line 157: “Transcripts with more than 10% of the coding sequence showing less than 5× coverage were considered to be absent from the transcriptome”. How to define coding sequence here? Need to be clear, not the entire transcripts are coding sequence.

Validity of the findings

1. Line 238: “None of the transcripts were entirely unique to either adults or juveniles, suggesting that the ontogenetic change is more quantitative than qualitative. Of the 17 putative toxin transcripts showing presence/absence variation” In Table 4 CTL-11 is one of the PAV transcripts, however, in Table 3 the TMP expression levels across ten assemblies, in ENP-A TMP 38.7 is viewed as absent, however, TMP 45 in LSG-A and TMP 47.1 in BR-A is viewed as present, so I wonder how much of these PAV transcripts is due to methods used in this study or due to un-equal amount of data in adult and juvenile, author need to provide more evidence the results is not due to other factors other than biological reasons.


2. Authors used three methods (outliers approach, DESeq, and DESeq2) to identify ontogenetic expression variation. 12 transcripts were identified overlap three methods, why not color code them in Figure 3. Currently in Figure 3 author listed only 17 differential transcripts by outliners approach, it would be good to use one color to indicate transcripts identified by all three methods (outliers approach, DESeq, and DESeq2) and one color indicated identified by (outliers approach, DESeq) and one color indicated identified by (outliers approach, DESeq2) and one color indicated by outliner approach only.


3. Good to see author used valid experiments to show the toxicity variation between adult and juvenile. I hope more genomics paper could do the similar type of experimental validation. Sometimes genomics change does not mean actual phenotype change, author did a nice job here!

---

## Round 0.2 · accepted · Accept

· Academic Editor

Accept

Many thanks for your responses. The only remaining change is that I would encourage (but not require) you to add in text explaining the rationale for pooling as explained in the response.